# Comparing Virtual and Center-Based Cardiac Rehabilitation on Changes in Frailty

**DOI:** 10.3390/ijerph20021554

**Published:** 2023-01-14

**Authors:** Evan MacEachern, Nicholas Giacomantonio, Olga Theou, Jack Quach, Wanda Firth, Ifedayo Abel-Adegbite, Dustin Scott Kehler

**Affiliations:** 1School of Physiotherapy, Dalhousie University, Halifax, NS B3H 4R2, Canada; 2Department of Cardiology, Dalhousie University, Halifax, NS B3H 4R2, Canada; 3Division of Geriatric Medicine, Dalhousie University, Halifax, NS B3H 4R2, Canada; 4Faculty of Health, Dalhousie University, Halifax, NS B3H 4R2, Canada; 5Hearts and Health in Motion, Nova Scotia Health, Halifax, NS B3L 0B7, Canada

**Keywords:** cardiac rehabilitation, virtual-care, frailty

## Abstract

Many patients with cardiovascular disease (CVD) are frail. Center-based cardiac rehabilitation (CR) can improve frailty; however, whether virtual CR provides similar frailty improvements has not been examined. To answer this question, we (1) compared the effect of virtual and accelerated center-based CR on frailty and (2) determined if admission frailty affected frailty change and CVD biomarkers. The virtual and accelerated center-based CR programs provided exercise and education on nutrition, medication, exercise safety, and CVD. Frailty was measured with a 65-item frailty index. The primary outcome, frailty change, was analyzed with a two-way mixed ANOVA. Simple slopes analysis determined whether admission frailty affected frailty and CVD biomarker change by CR model type. Our results showed that admission frailty was higher in center-based versus virtual participants. However, we observed no main effect of CR model on frailty change. Results also revealed that participants who were frailer at CR admission observed greater frailty improvements and reductions in triglyceride and cholesterol levels when completing virtual versus accelerated center-based CR. Even though both program models did not change frailty, higher admission frailty was associated with greater frailty reductions and change to some CVD biomarkers in virtual CR.

## 1. Introduction

Cardiovascular diseases (CVDs) are among the leading causes of hospitalization and mortality [1]. CVD disproportionately impacts older adults [2], who are likely contend with co-occurring health problems that impact their adverse outcome risk, compared to younger people [3]. Frailty describes the degree to which people accumulate these health problems with age, which results from decreased physiological reserve across multiple physiological systems that increases vulnerability to worsening health [4]. Evidence suggests a bi-directional association between CVD and frailty, as they share underlying physiological processes that increase the expression of one another [4,5]. Patients with more severe CVD are generally frailer [6,7,8], and frail CVD patients experience a greater risk of mortality compared to people with CVD and lower degrees of frailty [5,9]. 

Agencies that provide guidance on cardiovascular care have sought to mitigate the combined impact of frailty and CVD through cardiac rehabilitation (CR) [10]. CR is a comprehensive program for the secondary prevention of CVD [10] and is also effective for the improvement in frailty of participants [11,12,13,14,15]. CR implements behaviour change therapy consisting of nutritional guidance, medication management, CVD education, and exercise therapy to manage CVD in hospital settings, out-patient clinics, and alternatively, as virtual care [16,17]. Virtual CR is a home-based modification of traditional CR and is facilitated using the internet, telephone, or ‘smart-devices’ (e.g., smartphones, tablets) to remotely monitor progress and facilitate patient counseling [18]. Virtual CR has grown in popularity due to reduced center-based opportunities since the COVID-19 pandemic. Virtual CR shows similar improvements to center-based CR in managing cardiovascular biomarkers (e.g., cholesterol) [16], exercise outcomes (e.g., VO2 peak) [19], and quality of life for people [20] with a low-moderate CVD risk [21,22]. While virtual CR provides an opportunity to reach more people who could benefit from CR, little is known about the effect virtual CR has on frailty levels in CVD patients. Here, our objectives were to (1) compare the changes in frailty levels from CR admission to completion in patients who enrolled in either center-based CR or virtual-based CR, and (2) determine if admission frailty affects frailty changes and cardiovascular risk factors in both program models. 

## 2. Materials and Methods

### 2.1. Study Design

This study included 317 CR participants from the Hearts and Health in Motion CR program in Halifax, Nova Scotia, from August 2021–January 2022. Included participants were referred to CR following an acute adverse cardiovascular event by an automated referral system (i.e., following cardiac surgery) or healthcare professional (e.g., a cardiologist). The Nova Scotia Health Research Ethics Board approved this study. Eligible participants were adults 18-years of age or older who were referred and enrolled in CR for the secondary prevention of CVD. Participants were excluded if they withdrew from CR, cancelled participation for medical (e.g., critical illness) or personal (e.g., delayed enrollment) reasons, gave no response to frailty questionnaires at either CR admission or completion, or did not have an email address. 

### 2.2. Cardiac Rehabilitation

Prior to enrollment in CR, eligible participants were allocated to either virtual or center-based CR as determined by the multidisciplinary CR staff (Appendix A includes program details). When deciding, CR staff members primarily considered participants’ required level of supervision deemed necessary based on the participant’s health status at CR admission. To a lesser extent, CR staff also considered participants’ preference of program model to avoid subsequent participant drop-out. All CR participants performed baseline graded exercise stress testing for exercise prescription and safety. Participants deemed “low-to-moderate risk” (e.g., fewer mobility limitations) were preferentially allocated to the virtual CR program. The center-based CR program included “low, moderate, and high-risk” participants. 

The center-based CR program was a group-based, accelerated 6 week program offered from August–November of 2021, as was routine care during this period. The traditional 12 week center-based CR program was unavailable due to COVID-19 restrictions enforced by public health authorities in the region. Exercise sessions were supervised by a nurse and physiotherapist who measured exercise adherence by CR attendance. Exercise classes occurred once a week for 6 weeks, at a duration of 60 min per session (40 min exercise), including a warm-up and cool-down (20 min). Exercise types were continuous or interval aerobic exercise on a treadmill, or a leg or arm cycle ergometer (20 min each). Participants were encouraged to exercise at a self-monitored, moderate intensity of 11–13 on the Borg Rating of Perceived Exertion (RPE) scale. Exercise was progressed by increasing treadmill speed or incline, or ergometer resistance, while maintaining revolution speed. Center-based CR participants were also encouraged by CR physiotherapists to supplement weekly exercise classes with home-based exercise (e.g., walking), working in a stepwise fashion to meet Canada’s recommended physical activity guidelines of 150 min of moderate–vigorous exercise per week. Group-based education with CR staff provided information on how to manage CVD risk factors through health behaviour changes to diet, exercise safety, and medication management, if needed. Education included up to three weekly phone or Zoom video call rotations with the physiotherapist, nurse, and dietitian, supplemented by in-person consultations during exercise sessions. In total, center-based participants were eligible to receive up to nine hours of education time with CR staff. Center-based CR was delivered as a planned accelerated program.

Virtual-based CR participants received up to 10 weeks of individualized, unsupervised programming at to be completed at home. Physiotherapists prescribed 150 min of moderate–vigorous exercise to be completed weekly. Prescribed exercise plans in virtual CR were individually based on the resources each participant had access to (e.g., a neighborhood walk, body-weight exercises, or a treadmill at home). The type of exercises prescribed in the virtual program followed the same format as the center-based program, for example, continuous or interval exercise. Exercise intensity was consistent with center-based CR (RPE 11–13). Weekly education included up to four group-based Zoom video calls, and up to six individual telephone consultation calls, rotating between the physiotherapist, nurse, and dietitian. Here, physiotherapists recorded adherence, progressions, or modifications to the prescribed exercise. In total, virtual CR participants were eligible to receive up to 8.5 h of education time with CR staff. Virtual CR was subject to interruption and modifications due to COVID-19, detailed under limitations. Neither CR program reported an adverse event.

### 2.3. Frailty Index

A 65-item frailty index (FI) based on the Canadian Longitudinal Study on Aging (CLSA-FI) data was used to identify frailty at CR admission and completion (Appendix B, Appendix A). The CLSA-FI was developed in accordance with previous guidelines [23] and has been validated elsewhere [24]. CLSA-FI variables included signs, symptoms, diseases, and disability [24]. The presence of health deficits, such as diseases, were scored as 0 (deficit not present) to 1 (deficit present). Variables with three or more possible outcomes were scored on a grading scale from least to most severe based on the number of outcomes. The CLSA-FI is a ratio of the health deficits present divided by the total number of health deficits assessed to assign a score ranging from 0–1 (e.g., 20/65 = 0.31). Previous research has determined a small but clinically meaningful change using an FI equal to 0.03 [25,26]. Higher CLSA-FI scores indicate higher frailty levels. The FI has previously evaluated frailty level changes among CR participants [11]. We also developed an FI for sensitivity analyses by adding 8 cardiovascular biomarkers (described below) to the CLSA-FI (FI-CVD; Appendix A).

### 2.4. Cardiovascular Outcomes

Cardiovascular biomarkers included triglycerides, total cholesterol, HDL-cholesterol, LDL-cholesterol, creatine kinase, creatinine, c-reactive protein, systolic blood pressure, diastolic blood pressure, and resting pulse. Biomarkers were routinely collected in both CR models by CR staff, or through blood requisition at admission and upon completion. FI-CVD did not include creatine kinase and c-reactive protein, as CR staff advised confounding factors (e.g., medication changes, illness) may have influenced patients’ values over the course of CR. 

### 2.5. Statistical Analyses

Analyses were performed with R 4.1.3 (RStudio, Boston, MA) and SPSS Version 27 (IBM Corp, Armonk, NY, USA) software. Independent t-tests and Chi-squared tests compared differences in continuous and categorical descriptors of CR program models, respectively. A two-way mixed measures analysis of variance (ANOVA) examined frailty change from CR admission to completion in virtual versus center-based CR participants. Follow-up simple slope analyses centered FI scores from 0.05–0.25 because a pre-planned analysis revealed an interaction effect between admission frailty and CR program model on frailty change. Linear regression models were used to predict changes in cardiovascular biomarkers from admission CLSA-FI scores, stratifying by CR program model. All models were adjusted for exercise attendance and admission age, sex, triglycerides, total cholesterol, HDL cholesterol, LDL cholesterol, creatine kinase, creatinine, c-reactive protein, systolic blood pressure, diastolic blood pressure, and resting pulse. The “MICE” (Multiple Imputation Chained Equations) package was used to perform multiple imputation analyses to account for missing CLSA-FI and cardiovascular biomarkers. MICE imputed 1353/3407 (28.4%) missing data points on frailty and cardiovascular biomarkers, generating 100 predictive mean matching sequences. Little’s test determined our data was missing completely at random (Chi-squared = 836.634, degrees of freedom = 965, and *p* = 0.999). A two-sided *p*-value of <0.05 was considered statistically significant for all analyses. FI values were multiplied by 100 to improve the interpretability of findings. The individual who analyzed this study’s outcomes of interest was blinded to CR treatment allocation. 

### 2.6. Sensitivity Analyses

We completed two sensitivity analyses. First, we used the FICVD to measure change in frailty and CVD biomarkers. Second, we performed analyses using listwise deletion, whereby only participants with complete frailty data at admission and follow up were included. Frequency of individual CLSA-FI items from our listwise deletion CR participants are found in Appendix A. 

## 3. Results

### 3.1. Description of Participants

Three hundred and seventeen participants were screened for study inclusion (Figure 1). These participants were allocated to center-based (n = 165) and virtual CR (n = 152) programs. Of these 317 participants, 11 were excluded for primary prevention, one personal and five medical cancellations, five with no email address, and two with scheduling conflicts. An additional 24 withdrew from CR, and 137 did not respond to frailty assessments. The remaining 132 participants (mean age 64.5 ± 10.5, range 40–90, and 63.6% male) were enrolled in to virtual (n = 58) or center-based (n = 74) CR. 

Center- and virtual-based participants did not differ by sex, age, unadjusted mean admission CLSA-FI score, exercise attendance, or smoking status. A greater proportion of center-based participants had a history of stable coronary artery disease, while virtual participants were more likely to have coronary artery bypass graft surgery (*p* = 0.004) and hyperlipidemia (*p* = 0.018) (Table 1). 

### 3.2. Change in Frailty between Virtual and Center-Based CR

Admission and follow-up CLSA-FI scores after covariate adjustment were significantly higher in the center-based versus virtual CR program (Table 1; Figure 2A). However, frailty scores did not significantly change over time in either program model (F(116,1) = 0.477, *p* = 0.491). 

Subsequently, we conducted a sensitivity analysis by adding 8 cardiovascular biomarkers to the CLSA-FI (FICVD); we found frailty scores were slightly higher (Center-based: 0.159 vs. 0.146; virtual: 0.084 vs. 0.077) in both groups at admission (Figure 2B, Appendix A). Center-based participants had higher frailty scores with the FICVD at admission and completion, and both groups did not change their level of frailty after completing CR (F(116,1) = 0.746, *p* = 0.491). 

We examined a complete case sensitivity analysis by removing imputed data from our main analysis using listwise deletion. We found that listwise deletion CLSA-FI scores were significantly higher in center-based versus virtual CR participants, and frailty change was significantly different between CR models (F(51,1) = 11.873, *p* = 0.001; Appendix A). From admission to completion, center-based participants saw a significant CLSA-FI reduction of 0.016 (*p* = 0.018), while virtual participants saw a non-significant CLSA-FI increase of 0.006 (Appendix A). 

Simple slopes analysis revealed a significant interaction between admission frailty and CR model on frailty change (F(118,16) = 4.709, *p* = 0.002; Appendix A). We observed at low levels of admission frailty (CLSA-FI = 0.05) that frailty levels were significantly increased in virtual CR, relative to center-based CR, following program completion. Frailty did not differ between CR models for CLSA-FI scores centered at 0.10 and 0.15. However, at mild–moderate frailty levels (CLSA-FI ≥ 0.20), virtual CR participants observed a greater frailty reduction compared to center-based counterparts (Figure 3A, Appendix A). For example, after centering virtual CR participants’ admission CLSA-FI scores at 0.20 and 0.25, we observed corresponding beta coefficients of −3.810 (95% CI: −7.369, −0.251, *p* = 0.034) and −6.285 (−11.181, −1.390, *p* = 0.011), respectively. 

Results from our FICVD sensitivity analysis were consistent with simple slope analyses using the CLSA-FI ((F(115,16) = 2.105, *p* = 0.014); Figure 3B, Appendix A) indicating that mild–moderate frailty levels at admission were associated with greater frailty reductions in the virtual program compared to center-based CR; however, frailty did not increase at FICVD scores of 0.05. Our listwise deletion analysis found no significant interaction between admission frailty and CR model on frailty change ((F(50,16) = 1.603, *p* = 0.528); Appendix A).

### 3.3. Cardiovascular Biomarkers

We found no cardiovascular biomarker differences between CR models at admission; however, HDL-cholesterol was significantly higher in virtual participants at CR completion (Appendix A). Similarly, we found admission CLSA-FI was not predictive of change in cardiovascular biomarkers (Appendix A), and admission FICVD was only predictive of increased diastolic blood pressure in the virtual compared to center-based CR group (Appendix A). Simple slope analyses revealed significance between group differences for triglycerides and total cholesterol, such that virtual participants with higher admission CLSA-FI and FICVD (FI range = 0.20–0.30) saw greater associated reductions compared to center-based counterparts (Table 2, Appendix A). Listwise deletion analyses revealed admission CLSA-FI was associated with increased LDL-cholesterol (β-coefficient: 0.051[0.004,0.098], *p* = 0.033; Appendix A), and that virtual CR participants significantly increased their HDL-cholesterol, LDL-cholesterol, creatine kinase, and diastolic blood pressure compared to center-based participants (Appendix A).

## 4. Discussion

Interest in CVD and frailty is growing as researchers seek to better understand the coexistence of these two health concerns [27]. Accordingly, we studied changes in frailty from CR admission to completion with center-based versus virtual CR, as they were routinely implemented during COVID-19. We identified four key findings. First, center-based participants were significantly frailer than virtual participants upon CR admission. Secondly, mean differences in frailty change were not significantly different between CR models in our main analysis. Thirdly, frailty change was influenced by admission frailty level and CR model, such that frailer participants at admission (FI ≥ 0.20) reduced their frailty to a greater extent in virtual versus center-based CR. Fourth, admission frailty was associated with a change in some, but not all, cardiovascular biomarkers in virtual CR only. Here, we demonstrate that virtual CR is a reasonable alternative when center-based CR is inaccessible, enabling eligible patients to receive CR and improve their health.

Center-based participants had significantly higher CLSA-FI and FICVD scores than virtual participants at admission (Figure 2, Appendix A). This was expected, as participants who were deemed “low-to-moderate risk” by CR staff at admission were preferentially allocated to virtual CR. However, frailty levels were lower in our center-based sample compared to previous reports [11]. The discrepancy may relate to FI item differences or hesitancy among “higher risk” patients to enrol in CR during COVID-19. For example, previous research [11] used a 25-item FI with a greater ratio of CVD biomarkers than the CLSA-FI used here. Indeed, we observed higher FICVD versus CLSA-FI scores in both program models (Figure 2, Appendix A), highlighting the contribution of CVD biomarkers on frailty among CR participants. We acknowledge participant safety remains a priority for unsupervised virtual CR programs [21,28]. Therefore, our results support previously published literature which identify virtual-based health interventions as safe for low-to-moderate risk participants [21,22,29]. Yet, we observed participants with mild to moderate frailty levels in virtual CR, and thus we agree with previous statements arguing for more research using virtual CR in “high-risk” participants [21]. 

We show that on average, frailty, as measured by the CLSA-FI and FICVD, was not significantly changed in both program models (Figure 2, Appendix A). We anticipated that both program models would result in a lower frailty level, at least amongst people entering center-based CR, based on previous literature [11,12,13,14]. Conversely, our listwise deletion analysis showed significant differences between CR models on frailty change, such that center-based participants observed a small significant decrease (FI reduction of 0.016), while virtual participants observed a small non-significant increase in frailty scores (FI increase of 0.006) from CR admission to completion (Appendix A). However, these differences were not considered a clinically meaningful change in frailty (FI threshold: ≥0.03) [25,26]. Other studies demonstrated center-based CR programs of longer duration were associated with improvements in frailty; however, each of those CR programs operated for a minimum of 12 weeks (range = 12–24 weeks) [11,12,13,14]. Specifically, Kehler and colleagues and Mudge et al. each observed clinically meaningful reductions in frailty (i.e., ≥0.03) over the course of a 12 week exercise and education CR program [11,12]. Similarly, Lutz et al. reported frailty improvements using the frailty phenotype among CR participants completing a 12 week phase II program [13]. However, the aforementioned studies were conducted prior to COVID-19. In our study, COVID-19 restrictions enforced capacity and duration limitations to address the high volume of eligible CR participants on the waitlist, resulting in abbreviated CR programs (i.e., center-based = 6 weeks; virtual = 9–10 weeks). It is possible the limited volume of CR was insufficient to obtain similar reductions in frailty as observed in previous studies. Although our study aligns with findings from Kimber et al. in 2018 [30], whereby frailty was not improved among CR completers, we propose the lack of frailty change may be the result of an abbreviated CR duration for center-based participants; indicating a requirement for a standardized 12 week program. Thus, further investigation on the magnitude of frailty change as part of a 12 week virtual CR intervention is warranted. 

Although we did not identify differences in frailty change between center-based and virtual CR participants in our main analysis, simple slope analyses revealed an influence of admission frailty (CLSA-FI and FICVD), where higher frailty levels were associated with a greater magnitude of frailty reduction in participants enrolled in virtual CR (Figure 3, Appendix A). These findings are supported by previous literature [11,12]. Importantly, we found virtual CR participants with mild frailty levels (FI ≥ 0.20) improved to a greater extent than center-based equivalents (Figure 3; Appendix A). Our sensitivity analysis evaluating FICVD change demonstrated results consistent with our main analysis, while our listwise deletion analysis revealed no significance between group differences (Appendix A). Despite using multiple imputation, our results need to be interpreted with caution due to our small sample size of frailer participants at admission (Table 1). 

Finally, other than an increase in HDL-cholesterol in virtual participants, we identified cardiovascular biomarkers were unchanged irrespective of CR model (Appendix A). Moreover, admission CLSA-FI was not a predictor of change in cardiovascular biomarkers, and FICVD was only associated with increased diastolic blood pressure (Appendix A, respectively). Our simple slope analyses found virtual CR participants with higher admission CLSA-FI and FICVD scores saw a greater reduction in triglycerides and total cholesterol over the course of CR compared to center-based CR (Table 2, Appendix A; Appendix A and Appendix A, respectively). However, these changes were not observed at lower levels of admission frailty (Table 2, Appendix A). Although our findings support previous work favoring virtual over center-based CR on changes in HDL cholesterol [31], triglycerides [31,32], and total cholesterol [33], we caution our results as virtual CR programs provided 3–4 additional weeks for resolution of acute CVD events, and, based on clinical judgement of CR staff, virtual participants were considered ‘lower risk’ than the center-based participants. 

### Limitations

Our study has limitations. First, the different durations of the virtual and center-based CR programs do not allow for a true comparison of CR treatments, limiting the generalizability of our findings. Furthermore, the duration of the virtual and center-based CR programs did not follow the North American guidelines of CR programs (≥12 weeks) [10]. However, modified CR durations were necessary to accommodate a high volume of patients who were on a waitlist when CR programs were delayed as a result of COVID-19 precautions and public health guidelines, which provided insight into the impact of accelerated CR programming, resulting in termination of the 6 week center-based model with a return to 12 week programming. Secondly, our study prioritized CR participant safety during the allocation of CR programs, thus lacking randomization and introducing inherent selection biases in design. The decision to prioritize participant safety was deemed essential during a time of uncertainty and unexpected illness; however, we encourage future research to investigate frailty in virtual and center-based CR by randomizing consenting participants. Thirdly, virtual CR lacked standardization across program enrollments. Depending on virtual CR participants’ time of enrollment, participants would have received different programs due to CR closures, staff redeployment, and program adjustments during COVID-19. The inconsistency from shaping CR to address patients’ needs while appreciating program interruptions among different virtual programs should be considered when interpreting our study’s results [21]. However, these challenges were anticipated nationwide [22]. Fourth, we used multiple imputation to generate 28.4% missing variable values; however, this level of missingness is appropriate within multiple imputation guidelines [34]. Lastly, certain CLSA-FI items could not be reversed. Thirty five out of 65 items were reversible (e.g., difficulty with activities of daily living), whereas 30 out of 65 variables could only be accumulated (e.g., chronic diseases). 

## 5. Conclusions

We demonstrate virtual CR is non-inferior to center-based CR on frailty change; however, frailty improvements were significantly greater in frailer virtual participants at CR admission. Admission CLSA-FI scores may also be suitable for predicting change in some cardiovascular biomarkers.

## Figures and Tables

**Figure 1 ijerph-20-01554-f001:**
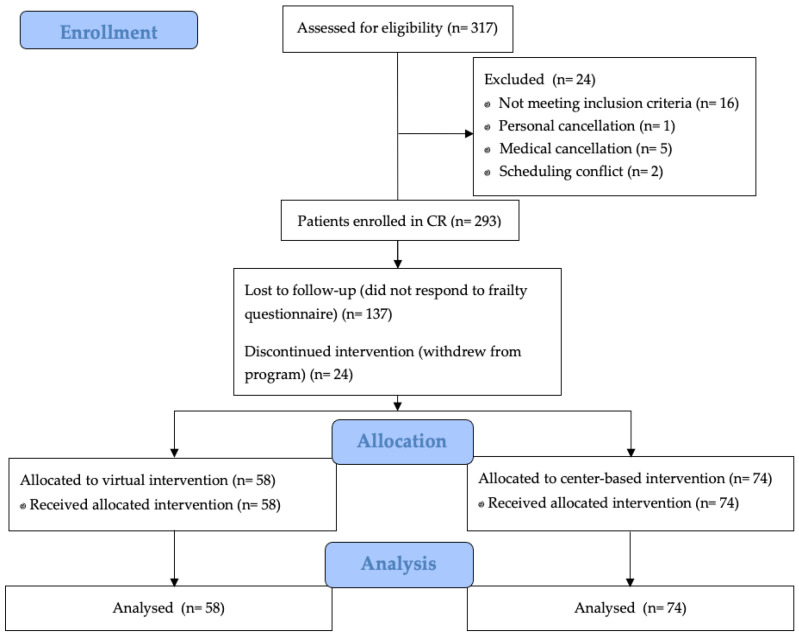
Flow diagram of study enrollment and CR program allocation.

**Figure 2 ijerph-20-01554-f002:**
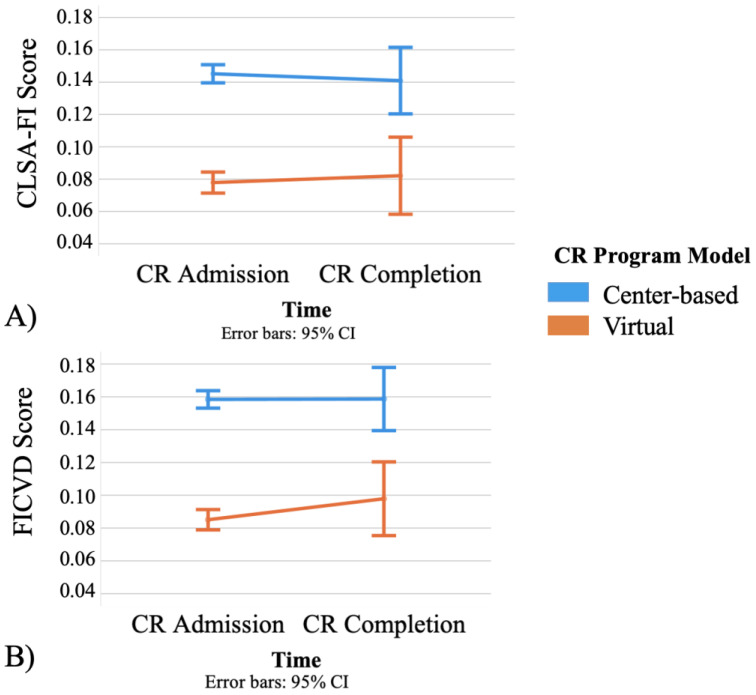
(**A**) Estimated marginal means of CSLA-FI frailty scores at admission and follow-up; (**B**) Estimated marginal means of FICVD frailty scores at admission and follow-up.

**Figure 3 ijerph-20-01554-f003:**
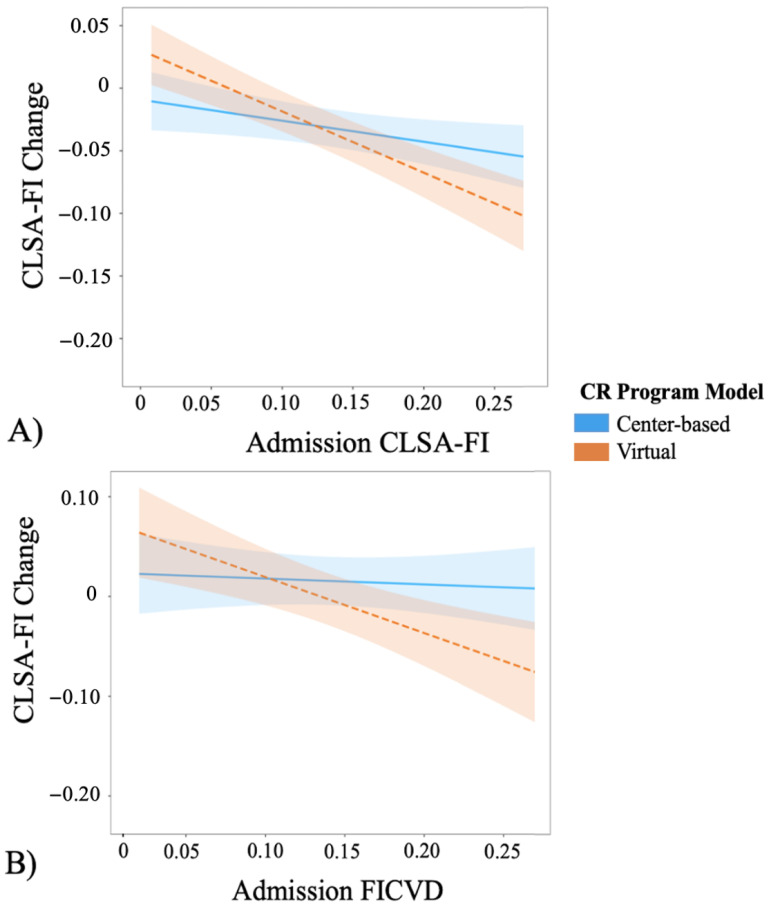
(**A**) Simple slope predicting CLSA-FI change by admission frailty, stratified by CR model; (**B**) Simple slope of FICVD predicting FICVD change by admission frailty, stratified by CR model. Shaded bands surrounding regression line represent beta 95% confidence intervals.

**Table 1 ijerph-20-01554-t001:** Demographic information of center-based and virtual cardiac rehabilitation participants at CR admission.

Variable	Cardiac Rehabilitation Model		*p* Value
	Center-Based	Virtual	
Sex - Male - Female	47 (63.5%) 27 (46.5%)	37 (63.7%) 21 (46.3%)	0.974
Mean age	63.1 ± 10.6	66.4 ± 10.1	0.069
Unadjusted admission CLSA-FI ^a^ - FI <0.10 - FI = 0.11–0.19 - FI = 0.20–0.29 - FI >0.30 **Adjusted admission CLSA-FI ^a,b^**	0.11 ± 0.07 35 (47.2%) 32 (43.2%) 5 (6.7%) 2 (2.7%) **0.14 ± 0.003**	0.11 ± 0.06 29 (50%) 24 (41.3%) 4 (6.8%) 1 (1.7%) **0.07 ± 0.003**	0.946 ***0.001 ****
Exercise session attendance	88.9% ± 17.9	88.9% ± 22.2	0.975
Cardiovascular biomarkers ^a^ - Triglycerides - Total cholesterol - HDL-cholesterol - LDL-cholesterol - Creatine kinase - Creatinine - C-Reactive protein - Systolic blood pressure - Diastolic blood pressure - Resting pulse	1.76 ± 1.01 3.74 ± 1.07 1.10 ± 0.28 1.85 ± 0.84 110.15 ± 64.48 86.65 ± 35.41 6.70 ± 16.57 122.72 ± 19.84 72.19 ± 10.11 66.89 ± 10.82	1.54 ± 0.76 3.43 ± 0.76 1.13 ± 0.24 1.59 ± 0.65 115.10 ± 75.66 77.28 ± 15.32 4.03 ± 5.25 125.53 ± 15.02 71.64 ± 9.05 65.45 ± 10.73	0.168 0.062 0.579 0.053 0.685 0.062 0.240 0.371 0.746 0.447
Smoking status - Current smoker - Former smoker - Never smoked - Missing	11 (14.8%) 34 (43.6%) 29 (39.1%) 2 (2.7%)	8 (13.8%) 24 (41.3%) 22 (37.9% 2 (3.4%)	0.884 0.603 0.863
History of CVDs ^a^ - **Stable coronary artery disease** - Acute coronary syndrome - Myocardial infarction - **Coronary artery bypass graft** - Cardiomyopathy - Percutaneous coronary intervention - Stroke	**19 (24.3%)** 9 (12.2%) 32 (43.2%) **4 (5.1%)** 3 (3.8%) 28 (37.8%) 3 (3.8%)	**6 (10.3%)** 5 (8.6%) 30 (51.7%) **17 (29.3%)** 2 (3.4%) 23 (39.7%) 1 (1.7%)	***0.026 **** 0.515 0.336 ***0.004 **** 0.858 0.833 0.442
CVD risk factors - Hypertension - **Hyperlipidemia** - Family history ^c^ - Diabetes - Inactivity - Obesity - Stress	58 (78.4%) **62 (83.8%)** 37 (50.0%) 22 (29.7%) 13 (17.5%) 13 (17.5%) 41 (55.4%)	44 (75.8%) **56 (96.5%)** 23 (39.7%) 16 (27.5%) 14 (24.1%) 6 (10.3%) 39 (67.2%)	0.734 ***0.018 **** 0.239 0.789 0.357 0.244 0.170

Data are presented as n (%) or mean ± SD from the multiple imputation dataset. ^a^ Abbreviations: CLSA-FI, Canadian Longitudinal Study on Aging Frailty Index; CVD(s), cardiovascular disease(s); HDL, high-density lipoprotein; LDL, low-density lipoprotein. ^b^ Adjusted variables include exercise attendance, admission age, sex, triglycerides, total cholesterol, HDL cholesterol, LDL cholesterol, creatine kinase, creatinine, c-reactive protein, systolic blood pressure, diastolic blood pressure, and resting pulse. ^c^ Family history included any history of coronary artery disease in immediate family: males < 55 years, females < 65 years. Computed at alpha = 0.05. Statistically significant values are listed in bold with corresponding *p* values listed in bold and italics.

**Table 2 ijerph-20-01554-t002:** Simple slope analyses of cardiovascular biomarker change by admission CLSA-FI^b^ and CR program model interaction.

Cardiovascular Biomarker	R Square	Beta 95% CI		F-Statistic	*p* Value
	Beta	Lower	Upper		
Simple slope analysis					
(Reference is center-based CR)					
Triglycerides - FI = 0.05 - FI = 0.10 - FI = 0.15 - FI = 0.20 - **FI = 0.25**	0.130 0.210 −0.033 −0.277 −0.522 **−0.766**	−0.001 −0.280 −0.392 −0.655 −1.053 **−1.508**	0.099 0.701 0.324 0.099 0.009 **−0.025**	*1.156* (116, 15)	0.054 0.392 0.851 0.143 0.051 ***0.040 ****
Total cholesterol - FI = 0.05 - FI = 0.10 - FI = 0.15 - **FI = 0.20** - **FI = 0.25**	**0.251** 0.408 0.051 −0.304 **−0.660** **−1.017**	**−0.125** −0.117 0.330 −0.706 **−1.229** **−1.811**	**−0.017** 0.933 0.433 0.097 **−0.092** **−0.222**	***2.602* (116, 15)**	***0.009 **** 0.123 0.786 0.132 ***0.021 **** ***0.011 ****
HDL-cholesterol ^a^ - FI = 0.05 - FI = 0.10 - FI = 0.15 - FI = 0.20 - FI = 0.25	0.390 −0.131 −0.108 −0.084 −0.061 −0.037	−0.015 −0.328 −0.252 −0.236 −0.275 −0.335	0.024 0.065 0.036 0.067 0.152 0.260	*4.951* (116, 15)	0.643 0.185 0.136 0.267 0.569 0.901
LDL-cholesterol ^a^ - FI = 0.05 - FI = 0.10 - FI = 0.15 - FI = 0.20 - FI = 0.25	0.178 −0.409 −0.354 −0.300 −0.246 −0.192	−0.054 −1.027 −0.806 −0.776 −0.918 −1.130	0.076 0.209 0.096 0.175 0.425 0.745	*1.683* (116, 15)	0.734 0.188 0.118 0.209 0.464 0.682
Creatine kinase - FI = 0.05 - FI = 0.10 - FI = 0.15 - FI = 0.20 - FI = 0.25	0.135 27.446 −14.774 −56.994 −99.215 −141.436	−18.739 −72.421 −87.629 −134.080 −208.194 −293.457	1.851 127.313 58.080 20.090 9.763 10.585	*1.212* (116, 15)	0.104 0.583 0.685 0.141 0.071 0.065
Creatinine - FI = 0.05 - FI = 0.10 - FI = 0.15 - FI = 0.20 - FI = 0.25	0.054 15.163 12.385 9.608 6.830 4.052	−10.558 −83.244 −60.246 −67.003 −100.248 −144.490	9.447 113.571 85.017 86.219 113.909 152.596	*0.447* (116, 15)	0.912 0.758 0.733 0.802 0.898 0.956
C-Reactive protein - FI = 0.05 - FI = 0.10 - FI = 0.15 - FI = 0.20 - FI = 0.25	0.254 −3.534 −3.215 −2.895 −2.577 −2.258	−1.374 −17.556 −13.471 −13.718 −17.822 −23.496	1.501 10.488 7.041 7.925 12.667 18.979	*2.638* (116, 15)	0.929 0.615 0.531 0.593 0.735 0.821
Systolic BP - FI = 0.05 - FI = 0.10 - FI = 0.15 - FI = 0.20 - FI = 0.25	0.185 −6.403 −5.105 −3.807 −2.509 −1.211	−0.749 −16.091 −12.238 −11.476 −13.355 −16.290	1.268 3.284 2.027 3.860 8.336 13.867	*1.760* (115, 16)	0.607 0.188 0.154 0.322 0.644 0.872
Diastolic BP - FI = 0.05 - FI = 0.10 - FI = 0.15 - FI = 0.20 - FI = 0.25	0.219 2.468 1.125 0.040 −1.173 −2.388	−0.835 −3.337 −2.975 −4.389 −7.410 −11.089	0.349 8.275 5.484 4.469 5.062 6.313	*2.174* (115, 16)	0.413 0.396 0.554 0.985 0.707 0.584
Resting pulse - FI = 0.05 - FI = 0.10 - FI = 0.15 - FI = 0.20 - FI = 0.25	0.152 0.125 −3.232 −6.589 −9.947 −13.305	−1.731 −10.291 −10.831 −14.534 −21.109 −28.867	0.387 10.543 4.367 1.354 1.213 2.256	*1.386* (115, 16)	0.207 0.980 0.396 0.099 0.077 0.089

^a^ CLSA-FI values were multiplied by 100 to increase interpretability of findings, corresponding beta-coefficients relate to 1-unit increases in CLSA-FI. Computed using alpha = 0.05. Model used center-based CR as the reference. Statistically significant values are listed in bold with corresponding *p* values listed in bold and italics.

## Data Availability

Restrictions apply to the availability of these data. Data was obtained from the Nova Scotia Health—Hearts and Health in Motion Cardiac Rehabilitation program and are available from the authors with the permission of Nova Scotia Health—Hearts and Health in Motion.

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
