# Peer review of "Comparing Virtual and Center-Based Cardiac Rehabilitation on Changes in Frailty"

_ijerph, 2023, doi:10.3390/ijerph20021554_

Round 1

Reviewer 1 Report

This is a paper that compares two different approaches to providing CR services  (virtual vs Center-based). Frailty scores as well as various markers were assessed. Admit frailty level and changes in frailty were also assessed.

Comments for author:

In a study with safety of patients being first priority, this does lead to some limitations in the study design that need to be both acknowledged but also incorporated into interpretation of results.

Example:

The exercise treatments were not the same. The virtual group exercised 150 min per week for 10 weeks. The Center-based group only exercised for 40 minutes, once per week, for 6 weeks. So one group had 1500 exercise minutes while another only 240 minutes. This makes true comparisons between CR treatments difficult for physical changes in function.

- The subject groups were not the same. While this is unavoidable for safety reasons, it makes inferences regarding the effectiveness of one CR method vs another because one group is different. Higher risk and lower frailty patients were placed in the center-based program.

Statistical analysis

This has me a bit confused. Initially it appears that there are no main effects for frailty, such that neither treatment showed changes in frailty. However a number of additional statistical tests were applied to suggest that frailty scores were different? (i.e lines 194-201

" Center-based participants had higher frailty scores with the FICVD at admission and completion, and both groups did not change their level of frailty after completing CR (F(116,1)=0.746, p=0.491).  Sensitivity analysis listwise deletion CLSA-FI scores were significantly higher in center- based versus virtual CR participants, and frailty change was significantly different be tween CR models (F(51,1)=11.873, p=0.001; Supplemental Table S4, Supplemental Figure 199 S1). From admission to completion, center-based participants saw a significant CLSA-FI 200 reduction of 0.016 (p=.018), while virtual participants saw a non-significant CLSA-FI in- 201 crease of 0.006"

Specific questions

- All subjects received a graded exercise test prior to placement. Do you have the MET values for these tests? 

- Home exercise often is part of a center based program. Did subjects do any home exercise?

- Table 1 What is "unadjusted admission" vs "adjusted admission"? I cannot find any explanation of this statistically significant  measure.

What is meaningful change in frailty? The figures showing pre-post frailty scores look essentially identical. Yet a larger portion of the paper is using different statistical modeling to suggest changes that may be present. Could the results simply be no change? Six weeks of CR with only 40 min per week of exercise does not seem to be a significant enough stimulus, which of course is why the full 12 weeks is so important. My point is that "no results" can still be informative results. 

- There are  tables with large volumes of non significant statistical information. These could be removed or significantly reduced in size (i.e table 2)

Discussion

This section should compare previous studies of longer length CR mentioned in the introduction. How did the present study compare? Could lack of change be the shortened time frame, indicating a need for the full 12 weeks? 

Author Response

Dear Blythe Liu, peer reviewers, and editorial team,

RE: Comparing Virtual and Center Based Cardiac Rehabilitation on Changes in Frailty

Thank you for the opportunity to revise and resubmit our original manuscript. We thank the reviewers for their valuable suggestions and opportunity to improve the original submission.

Please find below an itemized response to each reviewers’ comments as well as a revised manuscript (with tracked changes) where the revision is found.

Sincerely,

Scott Kehler

Reviewer comments:

Reviewer 1:

Comment 1: In a study with safety of patients being first priority, this does lead to some limitations in the study design that need to be both acknowledged but also incorporated into interpretation of results.

Response to comment 1:

We thank the reviewer for this comment. We have incorporated these points in the limitations section of the manuscript. Please refer to our responses below to address these comments.  

Comment 2: The exercise treatments were not the same. The virtual group exercised 150 min per week for 10 weeks. The Center-based group only exercised for 40 minutes, once per week, for 6 weeks. So one group had 1500 exercise minutes while another only 240 minutes. This makes true comparisons between CR treatments difficult for physical changes in function.

Response to comment 2:

This is an important part of the methods to clarify. COVID-19 posed challenges to the implementation of both virtual and center-based CR programs during the course of the study. Indeed, the durations of the programs were different due to an increasing waitlist of eligible CR participants. To resolve this issue, a truncated program was introduced and deemed necessary by the CR team. However, the volume of exercise between programs was matched, with a target of 150 minutes of moderate-vigorous exercise per week. Center-based participants supplemented their 60-minute exercise class (on-site) with home-based exercise to fulfill the remaining 90 minutes. We have added the following statements to section 2.2. Cardiac Rehabilitation, lines 94-97:

“Center-based CR participants were also encouraged by CR physiotherapists to supplement weekly exercise classes with home-based exercise (e.g., walking), working in a stepwise fashion to meet Canada’s recommended physical activity guidelines of 150-minutes of moderate-vigorous exercise per week”.

We have also added the following statements to lines 344-350:

“First, the different duration of the virtual and center-based CR programs do not allow for a true comparison of CR treatments, limiting the generalizability of our findings. Furthermore, the duration of the virtual and center-based CR programs did not follow the North American guidelines of CR programs (≥12-weeks). However, modified CR durations were necessary to accommodate a high volume of patients who were on a waitlist when CR programs were delayed as a result of COVID-19 precautions and public health guidelines…”

Additional clarification is provided in our response to comment 6. Please see Supplemental Table S1 for further description of CR program characteristics. 

Comment 3: The subject groups were not the same. While this is unavoidable for safety reasons, it makes inferences regarding the effectiveness of one CR method vs another because one group is different. Higher risk and lower frailty patients were placed in the center-based program.

Response to comment 3:

We agree with your comment that differences between groups. Our study and CR in Nova Scotia encountered unavoidable challenges associated with COVID-19. To provide acknowledgement, we have added the following statements to section 4.1. Limitations - lines 352-357:

“Second, our study prioritized CR participant safety during the allocation of CR programs, thus lacking randomization and introducing inherent selection biases in design. The decision to prioritize participant safety was deemed essential during a time of uncertainty and unexpected illness, however, we encourage future research to investigate frailty in virtual and center-based CR by randomizing consenting participants.”

Comment 4: Re: Statistical analysis.

This has me a bit confused. Initially it appears that there are no main effects for frailty, such that neither treatment showed changes in frailty.

However a number of additional statistical tests were applied to suggest that frailty scores were different? (i.e lines 194-201).

“Center-based participants had higher frailty scores with the FICVD at admission and completion, and both groups did not change their level of frailty after completing CR (F(116,1)=0.746, p=0.491). Sensitivity analysis listwise deletion CLSA-FI scores were significantly higher in center- based versus virtual CR participants, and frailty change was significantly different be tween CR models (F(51,1)=11.873, p=0.001; Supplemental Table S4, Supplemental Figure 199 S1). From admission to completion, center-based participants saw a significant CLSA-FI 200 reduction of 0.016 (p=.018), while virtual participants saw a non-significant CLSA-FI in- 201 crease of 0.006"

Response to comment 4:

We agree that this should be clarified. In our original submission, we supplemented our main analysis with two sensitivity analyses.

  1. Our main analysis found neither treatment (virtual nor center-based CR) resulted in significant changes in frailty over the course of the CR interventions.
  2. Sensitivity analysis 1 – Adding 8 CVD biomarkers to frailty index: Mean frailty index scores increased at both admission and follow-up timepoints in both groups, but still, no change in frailty over the intervention was observed.
  3. Sensitivity analysis 2 - Removed all imputed data by listwise deletion methods: Frailty change was significantly different between CR models. However, the listwise deletion sample was much smaller (67 participants; 25 virtual, 42 center-based) than our sample size from the main analysis (132 participants; 58 virtual, 74 center-based). Smaller sample sizes run the risk of over-estimation of effects, therefore, we spoke in most detail about the results of our main analysis with a greater sample size.

We hope that the reviewer is satisfied with this response

Comment 5: All subjects received a graded exercise test prior to placement. Do you have the MET values for these tests?

Response to comment 5:

Unfortunately, there was a large amount MET data from graded exercise testing and the level of missing data was too large to be eligible for multiple imputation. Due to challenges with COVID, we did not feel comfortable with providing this data. CR staff informed participants were hesitant to attend non-compulsory medical appointments, resulting in follow-up graded exercise stress tests not being completed in both the virtual and center-based programs. Baseline MET data was also sparse upon receipt of participants’ charts for data entry.

Comment 6: Home exercise often is part of a center-based program. Did subjects do any home exercise?

Response to comment 6:

Yes, thank you for requesting clarification on this. A description has been added to the section 2.2. Cardiac Rehabilitation – lines 94-97, as per our response to comment #2. Center-based participants were encouraged to follow Canada’s recommended guideline of 150 minutes of moderate-vigorous exercise per week, or rather, work in a step-wise fashion to reach this target over the duration of the program. Center-based participants were provided with physiotherapist consultations in the same manner as virtual participants to help facilitate home-based exercise.

Comment 7: - Table 1 What is "unadjusted admission" vs "adjusted admission"? I cannot find any explanation of this statistically significant measure.

Response to comment 7:

The adjusted admission CLSA-FI scores adjusted for cardiovascular biomarkers. These included exercise attendance and admission age, sex, triglycerides, total cholesterol, HDL cholesterol, LDL cholesterol, creatine kinase, creatinine, c-reactive protein, systolic blood pressure, diastolic blood pressure, and resting pulse. Please see section 2.5. Statistical Analysis – lines 151-154 for reference to the statement in our original submission:

“All models adjusted for exercise attendance and admission age, sex, triglycerides, total cholesterol, HDL cholesterol, LDL cholesterol, creatine kinase, creatinine, c-reactive protein, systolic blood pressure, diastolic blood pressure, and resting pulse.”

Comment 8: What is meaningful change in frailty? The figures showing pre-post frailty scores look essentially identical. Yet a larger portion of the paper is using different statistical modeling to suggest changes that may be present. Could the results simply be no change? Six weeks of CR with only 40 min per week of exercise does not seem to be a significant enough stimulus, which of course is why the full 12 weeks is so important. My point is that "no results" can still be informative results.

Response to comment 8:

As per previous work a clinically meaningful change in a frailty index is 0.031, 2.

In this study, we looked at frailty change as a mean difference between the two intervention types. We found that frailty was not significantly different pre-post in both interventions of CR. We indicated in our original submission that there was no change in frailty based on this analysis of frailty change by group over time (line 196-198; line 203-204).

“However, frailty scores did not significantly change over time in either program model (F(116,1)=0.477, p=.491).”

“…both groups did not change their level of frailty after completing CR (F(116,1)=0.746, p=0.491).”

Comment 9: There are tables with large volumes of non significant statistical information. These could be removed or significantly reduced in size (i.e table 2)

Response to comment 9:

We agree there is a lot of information in the tables, however, we believe it is important to show the estimates of effect based on our sample’s admission frailty level to better inform readers of our study. However, to improve the readability and interpretation of our results, we have bolded the statistically significant values in Table 1 and Table 2, and all corresponding Supplementary Tables.

Comment 10: RE Discussion. This section should compare previous studies of longer length CR mentioned in the introduction. How did the present study compare? Could lack of change be the shortened time frame, indicating a need for the full 12 weeks?

Response to comment 10:

Thank you for this comment. We have found that generally, frailty improvements in CR are more successful when the programs are of longer duration. Our discussion section appraises previous literature to contrast with the results of our study, as described on lines 298-315.

“Other studies demonstrated center-based CR programs of longer duration were associated with improvements in frailty, however, each of those CR programs operated for a minimum of 12-weeks (range = 12-24 weeks) [3-6]. Specifically, Kehler and colleagues and Mudge et al. each observed clinically meaningful reductions in frailty (i.e., ≥0.03) over the course of a 12-week exercise and education CR program [11-12]. Similarly, Lutz et al. reported frailty improvements using the frailty phenotype among CR participants completing a 12-week phase II program [13]. However, the aforementioned studies were conducted prior to COVID-19. In our study, COVID-19 restrictions enforced capacity and duration limitations to address the high volume of eligible CR participants on the waitlist, resulting in abbreviated CR programs (i.e., Center-based=6-weeks; Virtual=9-10-weeks). It is possible the limited volume of CR was insufficient to obtain similar reductions in frailty as observed in previous studies. Although our study aligns with findings from Kimber et al. in 2018 [7], whereby frailty was not improved among CR completers, we propose the lack of frailty change may be the result of an abbreviated CR duration for center-based participants; indicating a requirement for a standardized 12-week program. Thus, further investigation on the magnitude of frailty change as part of a 12-week virtual CR intervention is warranted.”

References:

  1. Jang, I. Y.; Jung, H. W.;  Lee, H. Y.;  Park, H.;  Lee, E.; Kim, D. H., Evaluation of Clinically Meaningful Changes in Measures of Frailty. J Gerontol A Biol Sci Med Sci 2020, 75 (6), 1143-1147; DOI: 10.1093/gerona/glaa003.
  2. Theou, O.; van der Valk, A. M.;  Godin, J.;  Andrew, M. K.;  McElhaney, J. E.;  McNeil, S. A.; Rockwood, K., Exploring Clinically Meaningful Changes for the Frailty Index in a Longitudinal Cohort of Hospitalized Older Patients. J Gerontol A Biol Sci Med Sci 2020, 75 (10), 1928-1934; DOI: 10.1093/gerona/glaa084.
  3. Lutz, A. H.; Delligatti, A.;  Allsup, K.;  Afilalo, J.; Forman, D. E., Cardiac Rehabilitation Is Associated With Improved Physical Function in Frail Older Adults With Cardiovascular Disease. Journal of cardiopulmonary rehabilitation and prevention 2020, 40 (5), 310-318; DOI: https://dx.doi.org/10.1097/HCR.0000000000000537.
  4. Mudge, A. M.; Pelecanos, A.; Adsett, J. A., Frailty implications for exercise participation and outcomes in patients with heart failure. Journal of the American Geriatrics Society 2021; DOI: https://dx.doi.org/10.1111/jgs.17145.
  5. Ushijima, A.; Morita, N.;  Hama, T.;  Yamamoto, A.;  Yoshimachi, F.;  Ikari, Y.; Kobayashi, Y., Effects of cardiac rehabilitation on physical function and exercise capacity in elderly cardiovascular patients with frailty. Journal of Cardiology 2020; DOI: 10.1016/j.jjcc.2020.11.012.
  6. Kehler, D. S.; Giacomantonio, N.;  Firth, W.;  Blanchard, C. M.;  Rockwood, K.; Theou, O., Association Between Cardiac Rehabilitation and Frailty. Can J Cardiol 2020, 36 (4), 482-489; DOI: 10.1016/j.cjca.2019.08.032.
  7. Kimber, D. E.; Kehler, D. S.;  Lytwyn, J.;  Boreskie, K. F.;  Jung, P.;  Alexander, B.;  Hiebert, B. M.;  Dubiel, C.;  Hamm, N. C.;  Stammers, A. N.;  Clarke, M.;  Fraser, C.;  Pedreira, B.;  Tangri, N.;  Hay, J. L.;  Arora, R. C.; Duhamel, T. A., Pre-Operative Frailty Status Is Associated with Cardiac Rehabilitation Completion: A Retrospective Cohort Study. Journal of clinical medicine 2018, 7 (12); DOI: https://dx.doi.org/10.3390/jcm7120560.

Reviewer 2 Report

'Comparing Virtual and Center-Based Cardiac Rehabilitation on Changes in Frailty' was an interesting and useful study that could be presented to patients by proving the effectiveness of cardiac rehabilitation programs in the current corona situation. Depending on the degree of FI, a simple slope analysis using a centered-based CR as a reference had been interesting.

What was the reason for discriminating the exercise of subjects who participated in virtual and centered-based programs?

Why was the length of the program offered different between the two groups? Have you considered the impact this might have on your results?

If the CR staff were involved in the subject selection, there was bias in the subject assignment. Did the CR staff provide only group-based education in both groups? Please describe so that we can rule out whether the CR staff played a role as a bias in the outcome variables reported by the two groups.

Please use the CONSORT Flow diagram to include all subjects who were included in the study and those who exited the study after the program.

Results and discussions are well presented.

Author Response

Dear Blythe Liu, peer reviewers, and editorial team,

RE: Comparing Virtual and Center Based Cardiac Rehabilitation on Changes in Frailty

Thank you for the opportunity to revise and resubmit our original manuscript. We thank the reviewers for their valuable suggestions and opportunity to improve the original submission.

Please find below an itemized response to each reviewers’ comments as well as a revised manuscript (with tracked changes) where the revision is found.

Sincerely,

Scott Kehler

Reviewer 2:

Comment 11: Comparing Virtual and Center-Based Cardiac Rehabilitation on Changes in Frailty' was an interesting and useful study that could be presented to patients by proving the effectiveness of cardiac rehabilitation programs in the current corona situation. Depending on the degree of FI, a simple slope analysis using a centered-based CR as a reference had been interesting.

Response to comment 11:

Thank you for this positive comment.

Comment 12: What was the reason for discriminating the exercise of subjects who participated in virtual and centered-based programs?

Response to comment 12:

We wanted to ensure that readers were aware of program differences with respect to achieving exercise volumes (150 minutes per week). Please refer to our response to comments #2 and #6 for further explanation.

Comment 13: Why was the length of the program offered different between the two groups? Have you considered the impact this might have on your results?

Response to comment 13:

Challenges associated with COVID-19 influenced the length of the two CR programs. The CR team wanted to reduce the growing waitlist number for enrolment, and therefore shortened the standard 12-week program to a 6-week or 10-week program to increase participant enrolment. We have acknowledged this is the 4.1. Limitations section, lines 348-352.

“However, modified CR durations were necessary to accommodate a high volume of patients who were on a waitlist when CR programs were delayed as a result of COVID-19 precautions and public health guidelines, which provided insight to the impact of accelerated CR programming, resulting in termination of the 6-week center-based model with a return to 12-week programming”.

Comment 14: If the CR staff were involved in the subject selection, there was bias in the subject assignment. Did the CR staff provide only group-based education in both groups? Please describe so that we can rule out whether the CR staff played a role as a bias in the outcome variables reported by the two groups.

Response to comment 14:

Thank you for identifying this. We now acknowledge selection biases in the 4.1. Limitations section, lines 352-357 (see below) and describe the allocation design in methods section under 2.2. Cardiac Rehabilitation, lines 73-82.

“Second, our study prioritized CR participant safety during the allocation of CR programs, thus lacking randomization and introducing inherent selection biases in design. The decision to prioritize participant safety was deemed essential during a time of uncertainty and unexpected illness, however, we encourage future research to investigate frailty in virtual and center-based CR by randomizing consenting participants.”

CR staff provided the same education materials to virtual and center-based participants. CR staff routinely collected CVD biomarkers including triglycerides, total cholesterol, HDL-cholesterol, LDL-cholesterol, creatine kinase, creatinine, c-reactive protein, systolic blood pressure, diastolic blood pressure, and resting pulse. However, CR staff did not have a role in the analysis or interpretation of these outcomes. The research team member who analyzed the outcomes was blinded to treatment allocation. We have added the following statements to section 2.5. Statistical Analyses, lines 161-162:

“The individual who analyzed this study’s outcomes of interest was blinded to CR treatment allocation”.

We have also added a footnote to Supplementary Table S2 & S3 describing the FI items and CVD biomarker items that were part of routine data collection.

Comment 15: Please use the CONSORT Flow diagram to include all subjects who were included in the study and those who exited the study after the program.

Response to comment 15:

Consort flow diagram is now provided.

Comment 16: Results and discussions are well presented.

Response to comment 16: Thank you for this positive comment. We appreciate it.

Round 2

Reviewer 1 Report

This reads better and more in line with the study. Given COVIG, data collection and group placement can be problematic, but there are still outcomes of interest to the reader.